# *Trypanosoma cruzi*: Genomic Diversity and Structure

**DOI:** 10.3390/pathogens14010061

**Published:** 2025-01-12

**Authors:** Alfonso Herreros-Cabello, Francisco Callejas-Hernández, Núria Gironès, Manuel Fresno

**Affiliations:** 1Centro de Biología Molecular Severo Ochoa, Consejo Superior de Investigaciones Científicas, Universidad Autónoma de Madrid, Cantoblanco, 28049 Madrid, Spain; 2Bloomberg School of Public Health, Johns Hopkins University, Baltimore, MD 21205, USA; bio.fcallejas@gmail.com; 3Instituto Sanitario de Investigación Princesa, 28006 Madrid, Spain

**Keywords:** *Trypanosoma cruzi*, strains, genomic structure, multi-gene families

## Abstract

*Trypanosoma cruzi* is the causative agent of Chagas disease, a neglected tropical disease, and one of the most important parasitic diseases worldwide. The first genome of *T. cruzi* was sequenced in 2005, and its complexity made assembly and annotation challenging. Nowadays, new sequencing methods have improved some strains’ genome sequence and annotation, revealing this parasite’s extensive genetic diversity and complexity. In this review, we examine the genetic diversity, the genomic structure, and the principal multi-gene families involved in the pathogenicity of *T. cruzi*. The *T. cruzi* genome sequence is divided into two compartments: the core (conserved) and the disruptive (variable in length and multicopy gene families among strains). The disruptive region has also been described as genome plasticity and plays a key role in the parasite survival and infection process. This region comprises several multi-gene families, including trans-sialidases, mucins, and mucin-associated surface proteins (MASPs). Trans-sialidases are the most prevalent genes in the genome with a key role in the infection process, while mucins and MASPs are also significant glycosylated proteins expressed on the parasite surface, essential for its biological functions, as host–parasite interaction, host cell invasion or protection against the host immune system, in both insect and mammalian stages. Collectively, in this review, some of the most recent advances in the structure and composition of the *T. cruzi* genome are reviewed.

## 1. Introduction to the Biology and Genetics of *Trypanosoma cruzi*

Recent advancements in the Trypanosomatidae family encompass a range of areas, including new understandings of trypanosomatid genetics and sexual processes, biodiversity, population structure, and host–parasite interactions [1,2,3,4,5]. Their ability to adapt to various environmental conditions and their high biological diversity allow these protists to significantly impact all biotic communities. The parasite *Trypanosoma cruzi (T. cruzi*) is responsible for Chagas disease, also known as American Trypanosomiasis, a chronic endemic, neglected tropical disease of Latin America.

The life cycle of *T. cruzi* is highly complex, involving four distinct stages in the parasite, an invertebrate hematophagous triatomine vector, and a broad range of mammalian hosts. In the midgut of the triatomine vector, the non-infective epimastigote stage replicates and then differentiates into the infective metacyclic trypomastigote in the hindgut. This infective form is then released in the feces or urine after a blood meal, allowing it to invade the host through mucosal surfaces or breaks in the skin. Once inside the host cells, *T. cruzi* escapes from the parasitophorous vacuole and transforms into the replicative amastigote. After several rounds of intracellular replication through binary fission, the amastigotes differentiate into trypomastigotes, which then lyse the host cell. The parasites enter the bloodstream, spreading the infection by invading other cells and tissues. Finally, when a triatomine takes a blood meal from an infected host, it ingests the bloodstream trypomastigotes, which then transform back into epimastigotes in the midgut of the vector, completing the life cycle [6].

*T. cruzi* is predominantly diploid and typically reproduces asexually through binary division, but there is evidence of natural hybridization, genetic exchange between strains, and sexual reproduction [7,8,9,10,11,12]. The population genetics of *T. cruzi* have sparked significant interest, leading to two contrasting views. One hypothesis, known as the clonal theory, suggests that *T. cruzi* exemplifies the predominant clonal evolution (PCE) model of pathogens, sharing many characteristics with other parasitic protozoa, fungi, and bacteria [13,14]. However, some researchers have shown that *T. cruzi* can reproduce sexually through a mechanism like classic meiosis, challenging the PCE model and suggesting it might not fully represent the biological complexity of this parasite [7,15], although it is important to highlight that the main form of reproduction of *T. cruzi* is clonal. Moreover, according to this clonal model, new genotypes have been assumed to develop through the gradual accumulation of distinct mutations, with minimal impact from rare genetic recombination events. Nevertheless, the observation of genetic exchange in vitro, the occurrence of sexual reproduction in nature, and the potential coexistence of clonal and sexually panmictic populations supports the existence of a sexual cycle in *T. cruzi* [9,16,17].

During mitosis, the genome of *T. cruzi* does not undergo condensation to form chromosomes, which hinders its visualization using conventional techniques [18,19]. Instead, the parasite karyotype has been determined using molecular biology techniques, such as pulsed-field gel electrophoresis (PFGE) combined with Southern blot analysis. These studies have unveiled significant molecular variability in both the size (0.45–4 Mb) and number (19–40) of chromosomes among strains and even within clones of the same strain [20].

Apart from the nuclear genome, trypanosomatids have an extra genome with the same levels of complexity as nuclear, defined by highly conserved and variable regions among strains. It is the mitochondrial or kinetoplast genome. Kinetoplastids possess a single large mitochondrion per cell [1] and their mitochondrial DNA forms a network of concatenated circular molecules known as the kinetoplast, consisting of maxicircles (~20–60 kb) and thousands of minicircles (0.5–10 kb) of varying sizes depending on the strains [21,22]. Maxicircles contain the mitochondrial genes found in other eukaryotes, and a divergent/variable region that can be more challenging to sequence due to its repetitive nature [23]. On the other hand, minicircles are unique to kinetoplastids and play a direct role in a U-insertion/deletion editing system by encoding guide RNAs (gRNAs) that are used for the maturation of the maxicircle transcripts [24].

## 2. Genomic Variability of *T. cruzi*

*T. cruzi* displays a very large genetic diversity. Nonetheless, there is no consensus on the heterogeneity of these populations. More than 6000 *T. cruzi* strains have been reported [25], and differences of up to 40% in the DNA content of clones of distinct strains have been detected, a variability that is primarily attributed to the nuclear DNA, with the highest recorded difference being 47.5% [26,27]. This wide variety of *T. cruzi* strains and the multitude of their genotypes have long been acknowledged due to their complex impact on the transmission cycles, epidemiology, and pathogenesis [28,29,30].

### 2.1. Genetic Classification of T. cruzi

Researchers have sought ways to categorize the *T. cruzi* strains primarily based on their biological and genomic differences, which gave rise to several classifications with different nomenclatures. We summarize here the most relevant classifications. In the 1990s, two main clades of *T. cruzi* (I and II) were identified based on biological, biochemical, and molecular criteria [31,32]. Subsequent analyses at the beginning of the XXI century, using gene and intergenic region sequences, suggested that this parasite can be classified into three major lineages, designated as clades A, B, and C [33], what was confirmed by another research that also described a fourth clade (D) and the presence of B–C hybrids [17]. Altogether, this data raised the possibility that *T. cruzi* could be a group of sub-species rather than just one. Furthermore, Brisse et al. proposed an alternative classification consisting of six phylogenetic lineages and proposed a change in the nomenclature of the initial *T. cruzi* groups, from I and II to I and IIa–e, based on phylogenetic information from multi-locus enzyme electrophoresis (MLEE) and random amplified polymorphic DNA (RAPD) markers [34].

Finally, after a Second Satellite Meeting in 2009, the six groups were numbered TcI to TcVI and termed as discrete typing units (DTUs) [35,36]. DTUs are defined as “sets of stocks that are genetically more related to each other than to any other stock and that are identifiable by common genetic, molecular or immunological markers” [37]. In this classification, TcV and TcVI represent hybrid lineages derived from haplotypes TcII and TcIII. Additionally, a seventh DTU initially restricted to bats (TcBat) was also described, closely related to TcI [38], and Flores-López et al. described by phylogenetic analysis a TcIV-USA lineage as a monophyletic group different to the South American TcIV and the other DTUs [39], emphasizing the fact that as additional vectors, hosts, and geographical areas are explored, we may discover and describe additional lineages.

However, some simplifications of these models have been proposed. First, in 2016, through the analysis of mitochondrial sequence genealogies, a new simplification of the DTU model was described. This analysis revealed three distinct mitochondrial clades showing a strong correlation with the previous DTUs: mtTcI corresponded to TcI, mtTcII to TcII, and mtTcIII included TcIII, TcIV, TcV, and TcVI [40]. Second, in 2021, another study performed a phylogenetic analysis using coding regions of maxicircles from up to 29 strains and 1108 single copy nuclear genes from all the DTUs, suggesting that *T*. *cruzi* is a complex of species composed of 2 major groups [41]. Group 1 would contain the previous lineages TcI, TcIII, and TcIV, and group 2 would contain the previous TcII, while hybrid strains (TcV and TcVI) would remain as a product of TcII and TcIII hybridization.

### 2.2. Strain Diversity

The first version of a *T. cruzi* genome, derived from the CL Brener strain, was published in 2005 [42], constituting a huge achievement for the molecular biology of the parasite. This strain was chosen to produce the *T. cruzi* genome reference given its high reproducible in vitro models of the acute phase, and a well characterized susceptibility to Benznidazole [43]. However, its hybrid nature was confirmed, and two haplotypes were identified to constitute the nuclear genome. These haplotypes were named according to their closest ancestor: Esmeraldo-like, derived from a TcII, and non-Esmeraldo-like, derived from a TcIII ancestor. Nonetheless, this assembly described for the first time the complex gene repertoire of *T. cruzi*. Unlike its closest counterparts *Leishmania major* and *T. brucei*, which have approximately 20–25% repetitive sequences in their genomes, *T. cruzi* exhibits around 50% repetition, complicating genome analysis and assembly [44].

Currently, multiple *T. cruzi* genomes are available in the National Center for Biotechnology Information (NCBI) database, facilitating the study of phenotypic, pathogenic, and complex variations among strains. Table 1 summarizes the available genomes for the different strains of *T. cruzi* with an associated publication, while Table 2 summarizes other available genomes that are not related to a research publication.

After the CL Brener assembly, the following sequenced strains were assembled above all with short-read sequencing methods (e.g., Illumina/Roche 454/Ion Torrent). This was the case for Sylvio X10/1 [45], Dm28c [46], Y [47], 231 [48], G, CL [49], and SC43 [50] strains of *T. cruzi* and the B7 strain of *T. cruzi marinkellei* [51]. Also, Gómez et al. sequenced six strains belonging to the different DTUs (except DTU TcIV) with Ion Torrent [52], and Reis-Cunha et al. assembled eleven TcII strains and two TcI strains with Illumina or Roche 454 [18,53]. Although these methods generate a high number of reads with low error rates, they are limited in their ability to produce complete chromosome reconstructions from short reads, leading to highly fragmented assemblies. This fragmentation can result in over-, under-, or misrepresentation of genes or entire chromosomal regions. Therefore, long-read sequencing methods (e.g., PacBio/PacBio HiFi/Nanopore) are considered more suitable for trypanosomatid genomes [54]. This technology enables the sequencing of long genetic fragments, circumventing the complex and repetitive nature of these genomes.

Long-read sequencing methods have the potential to produce genomes with a better resolution. Examples of strains assembled by these methods are Bug2148 [47], Dm28c, TCC [55], Tulahuen [56], and Dm25 [57]. However, these long-read methods exhibit a slightly higher error rate than short-read methods, being necessary to increase sequencing coverage (increasing the total cost per nucleotide) or combine technologies to minimize errors. Hybrid approaches have enhanced the assembly process, as demonstrated with the assembly of Berenice [58], Brazil A4, Y clone C6 [59], and STIB980 [60] strains.

Given the knowledge generated about genome assembly and genomic variability, we can conclude that more efforts are needed to complete as many genomes as possible to capture population dynamics and their evolution over time, especially for organisms, such as *T. cruzi*, whose complexity makes this process particularly difficult (and, therefore, expensive). More importantly, given the highly heterozygous nature and variable genomic repertoire between strains, we need as many genomes as strains described for *T. cruzi* for accurate transcriptomic and genomic analyses. Given the vast genomic variability and lack of synteny between strains (even classified into the same or close DTUs) [56], it is not possible to conclude that the latest genomes generated from hybrid assemblies and lower fragmentation levels can be used as universal genome references.

Considering the origin of the strains, the DTUs with more studied members are TcI and TcII, while to date there are no genomic assemblies for any TcIV strain. Notably, the origin of Bug2148 is under debate, since originally it was assigned to TcV, but some genomic and phylogenetic research has associated it with TcI [41,59,60,61], pointing out the possible mistake about the first assignation of this strain. Interestingly, Y and Dm28c strains have been assembled by different research groups and methods with very distinct results. While Y strain assemblies performed by short-read methods displayed sizes between 15 and 39 Mb, the combination of Illumina and Nanopore techniques gave 47 Mb. In the case of Dm28c, the difference is even more significant; short-read methods displayed sizes between 17 and 27 Mb, and the use of PacBio showed 53 Mb.

Transcriptomic data has also proven valuable for correcting and re-annotating previously assembled genomes. For instance, RNA-seq data improved the genome annotation of the Sylvio X10/1 strain, revealing that 79.95% of the genome corresponds to coding sequences, compared to the previous estimate of 37.73%. These findings suggest that the total genome size for Sylvio X10/1 is larger than previously reported, at least 51 Mb [62].

**Table 1 pathogens-14-00061-t001:** **Genome data of strains of *T. cruzi* and the B7 strain of *T. cruzi* marinkellei.** Contig N50: is a statistic median such that 50% of the whole assembly is contained in contigs equal to or larger than this value.

Strain	DTU	Size (Mb)	Contigs	Contig N50 (kb)	%GC	Date of Version	Sequencing Method	Reference
G	I	25.17	5531	6.70	50	11-2018	Roche 454	[49]
Dm28c	I	27.35	1210	78.39	50.5	11-2013		[46]
Dm28c	I	53.27	636	317.64	51.5	05-2018	PacBio	[55]
Dm28	I	17.23	6541	3.66	48.5	07-2021	Ion Torrent	[52]
B.M. López	I	18.51	5923	5.13	48.5	02-2020		
Sylvio X10/1	I	38.59	27,019	2.31	51	10-2012	Roche 454 + Illumina	[45]
STIB980	I	27.90	400	165.58	50.5	11-2023	Illumina + Nanopore	[60]
Brazil clone A4	I	45.56	697	191.35	51.5	11-2020	Illumina + PacBio	[59]
Dm25	I	45.40	179	496.17	51.5	02-2024	PacBio HiFi	[57]
Arequipa	I	19.05	10,332	1.91	51		Roche 454	[18]
Colombiana	I	30.85	9547	4.90	51			
S11	II	28.48	32,451	1.75	49	09-2018	Illumina	[53]
S154a	II	19.27	17,529	1.72	49			
S15	II	27.51	31,694	2.00	49			
S162	II	27.30	30,605	1.85	49			
S23b	II	28.13	32,315	1.87	49			
S44a	II	17.19	16,687	2.16	49			
S92a	II	27.08	31,256	1.91	49			
Y cl2	II	25.91	26,074	2.03	49			
Y cl4	II	26.14	26,957	2.06	49			
Y cl6	II	25.78	26,253	2.05	49			
Y nc	II	29.99	9164	5.13	50.5		Roche 454	[18]
Y	II	39.04	9821	11.96	50	10-2017	Illumina	[47]
Y	II	15.55	6942	2.89	50	07-2021	Ion Torrent	[52]
Y clone C6	II	47.22	477	396.94	51.5	11-2020	Illumina + PacBio	[59]
Berenice	II	40.80	934	148.96	51	06-2020	Illumina + Nanopore	[58]
Ikiakarora	III	18.49	11,096	2.19	48.5	02-2020	Ion Torrent	[52]
231	III	35.36	8469	5.30	48.6	01-2018	Illumina	[48]
SOL	V	20.06	11,944	2.17	49.5	07-2021	Ion Torrent	[52]
Bug2148	I/V	55.16	929	200.36	51.5	10-2017	PacBio	[47]
SC43	V	79.9	1318	238.74	51.5	11-2020	Illumina	[50]
CL	VI	26.77	6344	4.07	50.5	11-2018	Roche 454	[49]
TCC	VI	87.06	1236	264.20	51.5	05-2018	PacBio	[55]
Tulahuen	VI	48.46	75	872.48	52	12-2023	Nanopore	[56]
CL Brener	VI	19.53	11,101	2.3	49.5	07-2021	Ion Torrent	[52]
CL Brener	VI	89.94	32,746	14.67	51.5	08-2005	Sanger	[42]
B7	---	34.23	23,154	2.85	51	10-2012	Roche 454 + Illumina	[51]

**Table 2 pathogens-14-00061-t002:** **Genome data of different *T. cruzi* strains without an associated publication**. Contig N50: is a statistic median such that 50% of the whole assembly is contained in contigs equal to or larger than this value.

Strain	DTU	Size (Mb)	Contigs	Contig N50 (kb)	%GC	Date of Version	Sequencing Method
JR cl4	I	41.48	18,103	7.41	51.5	01-2013	Roche 454
Tula cl2	I	83.51	53,083	2.19	51.5	04-2013	Roche 454
Dm28c	I	50.93	1028	110.59	51.5	09-2017	PacBio
H1	I	27.34	11,257	16.44	49.5	02-2023	Illumina + PacBio + Nanopore
Esmeraldo cl3	II	38.08	20,187	5.35	51	01-2013	Roche 454

## 3. Genomic Structure

### 3.1. Chromosomes and Ploidy

Mitosis in *T. cruzi* occurs without a complete disassembly of the nuclear envelope. Although nucleosomes are present, chromatin does not condense into visible chromosomes, rendering classic cytogenetic studies unsuitable. Consequently, the karyotype of *T. cruzi* has been determined using molecular biology techniques, primarily PFGE in combination with Southern blotting [18,19,63,64]. Chromosomal band sizes range from 0.45 to 4 Mb, and the number of chromosomes has mainly been estimated by using probes as genetic markers. The number of chromosomes per haploid genome varies from 19 to 40, indicating that *T. cruzi* is predominantly diploid, although the sizes of homologous chromosomes as well as the chromosome lengths among clones of the same strain, strains from different DTUs, and even strains within the same DTU can differ significantly [18,20,65,66]. The analysis of ploidy or chromosomal copy number variation (CCNV) in this parasite was not feasible until the advent of the NGS methods.

The repetitive nature of the *T. cruzi* genome has led to difficulties in the genome assembly, making it challenging to determine the chromosome number and structure. Nevertheless, efforts to achieve full-length chromosome sequencing employed a combined strategy using synteny maps with *T. brucei* chromosomes and BAC end sequencing, resulting in the identification of 41 virtual chromosomes for the CL Brener strain [67]. Although the long-read methods overcame the high fragmentation limitations imposed by the short-read methods, producing contigs bigger than 1 Mb, likely covering entire chromosomes, some regions of the genomes remain fragmented. Therefore, the exact number and organization of chromosomes will ultimately be determined through the integration of long-read sequencing methods, optical mapping techniques, and polymer-based modeling, a field with huge advancements in the past decade [68].

*T. cruzi* is generally considered as a diploid organism. However, aneuploidies have been extensively studied in trypanosomatids, for example, in *Leishmania* spp., where ‘mosaic aneuploidies’ represent ploidy variations between isolates of the same strain and even among individual cells within the same population. These aneuploidies are associated with drug resistance, gene expression regulation, and host adaptation [69,70,71]. In contrast, *T. brucei* exhibits ploidy stability [72]. For *T. cruzi*, CCNV analysis depends on the quality of the assembled reference genome. Strains from different DTUs have shown that, like *Leishmania* spp., aneuploidy patterns vary both among and within strains and DTUs [18]. Despite this limitation, it has been observed that strains from DTU TcI appear to be more stable, whereas strains from DTUs TcII and TcIII exhibit a high degree of aneuploidies, including monosomies, trisomies, and tetrasomies [53].

These findings suggest that aneuploidy events in *T. cruzi* may facilitate gene expansion and alterations in gene expression, which is particularly crucial for parasites that rely on post-transcriptional mechanisms for gene regulation. While aneuploidies are generally associated with deleterious phenotypes in many eukaryotes, they may play a role in species-specific adaptations during trypanosomatid evolution. This could impact multi-gene families that are essential for establishing productive infections in mammalian hosts [73]. Regarding gene expansion processes, just a few mechanisms have been proposed. The main mechanism consists of homologous recombination at the disruptive compartment of the genome, mainly at the telomeric and sub-telomeric regions. This mechanism includes the participation of transposable elements’ nucleases, which introduce chromosome breakages, and the subsequent repair (by homologous recombination) introduces new gene copies, variants, and pseudogenes. This mechanism also considers (in a lesser proportion) the possibility of recombination between non-homologous chromosomes (ectopic recombination) [74,75,76,77].

### 3.2. Genome Organization

*T. cruzi* displays significant genomic plasticity and an unusual gene organization with tandemly repeated sequences, multi-gene families, and retrotransposons constituting over 50% of its genome. This genomic plasticity is linked to its genetic composition and is compartmentalized into two principal regions of protein-coding genes. The first is the core compartment, which contains highly conserved genes with known, putative, and constitutive functions, typically annotated as hypothetical conserved genes that exhibit synteny with other species, such as *Leishmania major* and *T. brucei*. The second is the non-syntenic disruptive strains-specific compartment, primarily formed by genes that evolve rapidly, including the surface multi-gene families [78,79].

Furthermore, the *T. cruzi* genome comprises three main types of DNA: first, repetitive sequences, including retrotransposons, tandem repeats, and short repeat elements; second, coding sequences of multi-copy gene families, such as those encoding surface proteins and virulence factors; third, coding sequences of single-copy genes that are conserved across strains and species, although the total number estimation of these single-copy genes differs between hybrid and non-hybrid strains [79]. Notably, approximately 50% of the genetic content has unknown functions [47], which aligns with proteomic studies of strains of different DTUs, where 40–50% of the total proteins are of unknown function [80,81,82,83]. This highlights the urgent need for studies to unravel these significant gaps in our understanding of *T. cruzi* genetics and biology.

The core and disruptive compartments exhibit variable %G+C content. Indeed, genes with high recombination rates and constant evolution tend to have elevated levels of G+C [84,85], which is correlated with the fact that the *T. cruzi* core compartment has a G+C content of ~48%, while the disruptive compartment has ~53%. Other studies also confirmed that variations in the G+C content are correlated with specific telomeric repeats in *T. cruzi*, such as the hexameric repeat TTAGGG and poly-T structures [75,86]. Precisely, the telomeric and sub-telomeric regions in this parasite are hotspots for DNA recombination, leading to extensive genetic variations and continuous evolutionary processes, affecting the relative abundance and gene organization [74,87].

### 3.3. Replication Origin

In eukaryotic organisms, chromosomes are replicated from numerous DNA replication origins (ORIs), ranging from hundreds to thousands, which are identified by the binding of the origin recognition complex (ORC). In *T. cruzi*, the ORIs of the CL Brener strain were mapped using marker frequency analysis sequencing (MFA-seq) [88], identifying 103 and 110 putative ORIs in each haplotype of this hybrid strain. The analysis revealed that some replication initiation sites are located at the boundaries of transcription units, as in *Leishmania major* and *T. brucei*. Notably, most of the predicted ORIs were abundant within coding DNA sequences and showed a high G+C content (averaging 65%), compared to the average of 54% along the genome. Additionally, DNA combing analysis of the same strain showed a median inter-origin distance of 171.1 kb (mean 208.3 ± 29.24 kb) [89].

While some ORIs in *T. cruzi* are found in non-transcribed regions, like those in *T. brucei* and *Leishmania major*, many are strategically positioned in sub-telomeric regions, particularly near disperse gene family 1 (DGF-1) genes, where they could generate genetic variability in multi-gene families [88]. The transcription orientation towards telomeres suggests that the high density of ORIs in sub-telomeric regions would lead to head-on collisions between transcription and replication processes, as replisomes move towards the chromosome centers. These findings indicate that such collisions are frequent, contributing to genetic variability, as evidenced by increased single nucleotide polymorphism (SNP) levels in sub-telomeric regions and DGF-1 genes containing putative ORIs.

### 3.4. Chromatin, Transcription and Gene Regulation of T. cruzi

Nucleus and chromatin are different depending on the *T. cruzi* stage. In epimastigotes and amastigotes, the nucleus is spherical with a large nucleolus and high quantities of euchromatin (transcriptionally active), while the trypomastigotes display an elongated nucleus, absence of nucleolus, and abundant heterochromatin (transcriptionally inactive), which is dispersed in the nucleoplasm [90,91,92]. Nuclear morphology alterations start at the onset of metacyclogenesis, with heterochromatin progressively spreading throughout the nucleus in intermediate forms. The precise mechanisms governing and regulating these ultrastructural changes during metacyclogenesis remain largely unknown [90]. Moreover, in a recent study evaluating the open chromatin status genome-wide, the core and disruptive genomic compartments of the *T. cruzi* genome exhibited huge differences in open chromatin enrichment. The core genomic region was highly enriched in open chromatin compared to the disruptive compartment [93].

In trypanosomatids, the organization of genes, gene expression regulation, and RNA metabolism display distinct characteristics compared to other eukaryotes. Genes are arranged into directional gene clusters (DGCs), which are separated by strand-switch regions (SSRs) where transcription directions either converge or diverge [94]. These DGCs are then transcribed as extensive polycistronic transcription units (PTUs), and the mRNA maturation involves co-transcriptional trans-splicing of a capped spliced leader (SL) RNA and polyadenylation [95]. Figure 1 shows this polycistronic transcription of *T. cruzi*.

Furthermore, it is widely recognized that post-transcriptional regulation plays a predominant role in gene expression regulation [96]. By modifying RNA transcripts after they are produced, *T. cruzi* can quickly respond to changes in its environment favoring its survival, and it can do it in a stage-specific manner, which is essential for its ability to adapt to different environments and hosts. Also, this process is particularly important because *T. cruzi* lacks the typical transcriptional control mechanisms found in other organisms [97]. Nevertheless, recent studies have underscored the significant influence of histone variants, histone post-translational modifications, base J (a hypermodified nucleobase of kinetoplastids), nucleosome positioning, and chromatin organization on gene expression regulation, cell cycle control, and differentiation [93,98,99,100].

Additionally, only a limited number of transcriptional regulators have been identified in trypanosomes, and the influence of genome spatial organization on gene expression remains poorly understood. A recent study mapped genome-wide chromatin interactions in *T. cruzi* using chromosome conformation capture (Hi-C), displaying that the core and disruptive regions create two 3D-chromatin compartments, named C and D. Both chromatin compartments presented distinctions in levels of nucleosome positioning, DNA methylation, and chromatin interactions, which produce a change in genome expression dynamics. The authors concluded that the trypanosome genome is structured into chromatin-folding domains, with transcription being influenced by the local chromatin architecture. They proposed a model suggesting that epigenetic mechanisms play a role in regulating gene expression in trypanosomes [94].

As previously noted, genes in *T. cruzi* are organized into non-overlapping clusters on the same DNA strand, often with unrelated predicted functions. These genes are transcribed as polycistronic units and subsequently undergo trans-splicing and polyadenylation. Although no shared consensus motifs or patterns have been identified among the SSRs, these are functionally active. For example, transcription initiation and termination occur within SSRs, and they are also implicated in DNA replication origin and centromeric functions [101,102,103]. Also, SSRs exhibit a distinct composition compared to the rest of the genome and different higher intrinsic curvatures [104], related to the transcriptional regulation. Notably, SSRs from the disruptive compartment are longer than those from the core compartment, with mean lengths of approximately 4.5 kb and 1.5 kb, respectively [78].

The mRNA maturation by trans-splicing is a unique RNA processing mechanism in which two exons from different genomic locations combine to form a single transcript [105]. In the particular case of *T. cruzi*, this involves inserting a 39-nucleotide sequence at the 5′ end of each transcript, known as the mini-exon or spliced leader (SL), which is transcribed from a tandem array as a precursor of approximately 140 nucleotides and undergoes capping modification. The insertion of this Cap-SL stabilizes the mRNA and facilitates the excision of each mRNA from the PTU, allowing for final polyadenylation [106]. Furthermore, the AG dinucleotide has been identified as the consensus sequence for SL trans-splicing in *T. cruzi* [62], *Leishmania major* [107], and *T. brucei* [108]. Nevertheless, slight variations in the nucleotide composition surrounding the AG dinucleotide suggest species-specific mechanisms for mRNA maturation.

Finally, the other well-known process for the mRNA maturation is polyadenylation. The typical AAUAAA polyadenylation signal found in eukaryotes is absent in trypanosomatids. A recent study showed that *T. cruzi* utilizes a single nucleotide as the most likely polyadenylation signal, with cytosine being the most frequent (45.3%) and thymine the least frequent (6.79%) [62]. This is distinct from other trypanosomatid species, such as *Leishmania major* and *T. brucei*, which use an AA dinucleotide as the most probable polyadenylation signal [107,108].

## 4. Multi-Gene Families of *T. cruzi*

*T. cruzi* harbors many multi-gene families, some comprising hundreds of members, which contribute to the repetitive nature of its genome. Extensive research has been conducted to elucidate the structure, distribution, and functions of these multi-gene families, which predominantly encode surface proteins that play crucial roles throughout the *T. cruzi* life cycle, from facilitating effective host–cell interactions and invasion to providing protection against the host immune system. Additionally, they exhibit significant expansion and continuous evolution, resulting in considerable diversity among strains [109].

Berná et al. identified multi-gene families, such as TSs, mucins, and MASPs, within the disruptive compartment of the *T. cruzi* genome, while retrotransposon hot spot proteins (RHS), GP63, and DGF-1 families were found in both disruptive and core compartments [55]. The specific reasons for this difference remain unknown. Copy numbers of these multi-gene families in the genomes of various *T. cruzi* strains are illustrated in Figure 2. Data indicates that the TS family is the most expanded multi-gene family, followed by MASPs and RHS, although this pattern is not consistent across all strains. For example, SylvioX10 presents a huge number of DGF-1 members in CL Brener but just a few RHSs. The high variability among strains may be attributed to strain-specific genetic profiles, the accuracy of the genome assemblies, and genomic plasticity. This diversity is supported by the observed differences in infection kinetics, virulence, and immune responses among *T. cruzi* strains [110,111].

### 4.1. Trans-Sialidases (TSs)

This multi-gene family is the largest of *T. cruzi*, playing a vital role in host–parasite interactions [112,113]. Their members are located on the membrane surfaces of metacyclic and bloodstream trypomastigotes, as well as intracellular amastigotes, and they are mainly distributed along the flagellum, cell body, and flagellar pocket [114]. Also, TSs have been found differently expressed among strains in extracellular vesicles [115]. Interestingly, the TS copy number is significantly smaller in *T. brucei* and is absent in *Leishmania major* [44].

These proteins display 2 structural motifs, a lectin-like motif plus a beta barrel motif, and they can present a glycosylphosphatidylinositol (GPI) anchor. However, it can be removed by the action of a phosphatidylinositol phospholipase C, allowing TSs to be released into the bloodstream. Also, some researchers have established that all members share a VTVxNVxLYNR canonical motif [116], although some TSs exhibit remarkable motif degeneration in compliance with recent analysis [79].

According to the genome annotation of the distinct strain assemblies of Figure 2, independently of the DTU of origin, all strains possess more than 1000 TS proteins. Notably, Bug2148, TCC, and Brazil A4 are the strains with more members, strains assembled by long-read sequencing methods or the combination of both long- and short-read methods, while CL Brener is the strain with least members, likely because it was not sequenced by these methods. Many TS genes are located near telomeric and sub-telomeric regions, which can cause assembly collapses and lead to under- or over-representation of these genes [75]. This supports the use of both long- and short-read methods for the assembly of *T. cruzi* genomes and the fact that there may still be errors in estimates of the real number of TS members in some parasite genomes. Also, this suggests that TS expansion is partly due to their chromosomal location and the selective pressure exerted by the host immune system, as TS proteins are targets of both humoral and cell-mediated immune responses [117].

Regarding the biological function of this family, the best-characterized function is the trans-sialidase catalytic activity, first described in 1980 [118]. Subsequent studies have demonstrated that *T. cruzi* is unable to synthesize its own sialic acids and utilizes TS enzymes to incorporate sialic acids from host cell sialoglycoconjugates into acceptor molecules on their membranes [79]. This sialylation process confers a negatively charged coat that protects trypomastigotes from being targeted by human anti-α-galactosyl antibodies and the complement system [119,120]. Further research suggests that TS activity is crucial for *T. cruzi* survival and the establishment of an effective infection [120]. Notably, although neuraminidase activity was identified in TS enzymes, it is only active in the presence of suitable Gal acceptors [121].

Nevertheless, the critical residues necessary for catalytic activity have been identified in only a few genes, and additional roles related to host–ligand interactions and immune regulation have been proposed [116,122]. Indeed, TS enzymes interact with various mammalian host cells, including neurons, Schwann cells, cardiac fibroblasts, B cells, thymocytes, CD4+ and CD8+ T cells, platelets, and endothelial cells [123], although many of their ligands remain unknown. Therefore, given that not all members of this protein family seem to exhibit TS activity, it may be advisable to consider renaming the family to better reflect its diverse functional characteristics, and more studies are mandatory to unravel the exact function of each member in the family.

Regarding the classification of the TSs, the last aggrupation was proposed in 2011 [122]. This classification was established by a sequence cluster analysis with the CL Brener strain separating the TSs in 8 groups. Groups II and V displayed the largest number of members, accounting for approximately 70% of the TSs analyzed. However, this study had two very important limitations considering the genome complexity of this parasite: it was done with the annotated proteins from just one strain, CL Brener, and its genome proceeds from a non-NGS assembly, which is not advisable for genomes with highly repetitive sequences, like *T. cruzi*. Therefore, this classification may be very far from reality.

Even so, different research groups have classified the trans-sialidases detected in their assemblies according to this 8-group classification. Figure 3 represents the distribution of the groups among these strains, considering only these annotated trans-sialidases. Group V is the predominant TS group, constituting approximately 50% of the TSs in each strain, except for Brazil A4, in which it accounts for 55%. In contrast, groups I, III, IV, and VII are significantly smaller. The overall profile is consistent across strains, although some exhibit notable variations. For instance, the Y and Bug2148 strains have over 10% of group I TSs, whereas other strains have less than 4%. Also, certain strains lack specific groups almost entirely, such as Brazil A4 with group III, Y with group IV, and Dm28c with group VII. Regarding group VI TSs, the TCC, Dm28c, and Brazil A4 strains display more than 12%, while other strains have less than 8%.

### 4.2. Mucin-Associated Surface Proteins (MASPs)

MASPs form the second largest multi-gene family of *T. cruzi*. They receive that name from their cluster position among mucin gene groups. Although both proteins differ in sequence, they are structurally similar [124]. MASPs display conserved N- and C-terminal domains encoding a signal peptide and a GPI-anchor addition site, respectively. These proteins are GPI-anchored to the membrane and predominantly expressed during the bloodstream trypomastigote stage [109], although they are also expressed in amastigotes and epimastigotes [80,125]. Moreover, MASPs are differentially expressed in clones from the same original population [126], and previous studies have demonstrated that their expression varies based on the type of host cell infected and across successive passages in mouse models [127]. The central region of MASPs varies in both sequence and length (176–645 amino acids), containing numerous repetitive motifs. Single amino acid repeats, particularly those with glutamic acid, are common, comprising about 27% of the identified motifs, while at least four potential O-glycosylation sites per sequence have been identified, with 70% of these sites being threonines [124].

Regarding the functions of the MASPs members, the specific roles in the parasite life cycle remain largely unclear. However, some researchers have begun to shed light on certain functions. The overexpression of MASPs in intracellular parasites, before amastigote division, suggests that these proteins play a crucial role in the survival and multiplication of intracellular amastigotes [124,128]. Additionally, MASP49 has been proposed as a virulence factor of *T. cruzi* [129], and the high degree of polymorphisms in the MASPs along with its location at the surface of trypomastigotes strongly suggests participation in host–parasite interactions [130]. In fact, MASP proteins have been demonstrated to stimulate antibody production during the acute infection stage [127]. Consequently, *T. cruzi* could express these surface proteins to produce unspecific T cell and antibody responses, facilitating parasite persistence within the mammalian hosts due to the variations in the extensive range of antigenic peptides within the MASP family. Furthermore, the N-terminal motif of MASPs has been shown to partially inhibit complement-mediated lysis [131], and immature MASPs, as well as the C-terminal portion, may be included in trypomastigote exosomes and exhibit immunogenic properties [131,132].

Considering the possible classifications of this multi-gene family, MASP members were divided into 7 subgroups (M1–M7) in recent research using hierarchical clustering analysis based on protein sequence similarity [130]. However, as is the case for the TS family, this classification was proposed just with the sequences of the CL Brener strain, hence a revision is needed. Figure 4 shows the abundance of each subgroup. Subgroups M1–M4 are the largest, while M5–M7 represent the smallest ones. Interestingly, in this study, the authors examined the antigenicity of these MASP subgroups, revealing that the recognition profiles are distinct between them. These profiles tend to vary during the acute infection stage and after multiple consecutive passages in murine models.

### 4.3. Mucins

This multi-gene family is the most abundantly expressed in the membrane of *T. cruzi* and ranks as the fourth largest multi-gene family [133]. Mucins display a GPI-anchor, a dense array of O-linked oligosaccharides attached to serine and/or threonine residues, and they are the main acceptors of sialic acid in the parasite membrane [134]. Moreover, they are widely distributed over the cell body, flagellum, and flagellar pocket of the different *T. cruzi* stages [135].

Regarding their role in the infection process, two primary functions have been associated with mucins. First, they protect the parasite from host defense mechanisms, supporting the persistence of *T. cruzi* [109]. For example, epitopes in mucins generate strong but delayed and non-protective CD8+ T cell responses [136]. Also, mucins members are expressed in large amounts in all the parasite stages conferring high resistance against proteases and glycosidases, and their sialylation by TSs protects against the complement-independent lysis induced by human anti-α-galactosyl antibodies [133]. Second, mucins facilitate the attachment and invasion of specific host cells [137].

Mucins are categorized into two subfamilies, TcMUC and TcSMUG, based on structural and biological criteria (Figure 5). TcMUC proteins are exclusively expressed during the mammalian stages of the parasite (amastigotes and bloodstream trypomastigotes), whereas TcSMUG proteins are found in the insect-dwelling forms (epimastigotes and metacyclic trypomastigotes) [138]. The greater diversity observed in TcMUC proteins compared to TcSMUG proteins is linked to the immune system pressures they encounter in mammalian hosts [137] and their chromosomal localization near telomeric regions [47]. Furthermore, TcMUC proteins feature a signal peptide, a GPI anchor, and a central region rich in threonines that are targets for O-glycosylation and subsequent sialylation, which may explain the higher glycosylation levels observed in mucins of amastigotes and bloodstream trypomastigotes compared to those in epimastigotes [137].

TcMUC genes are subclassified into TcMUC I, II, and III. TcMUC I genes are more abundant in amastigotes, while TcMUC II genes are predominantly found in the membrane lipid rafts of bloodstream trypomastigotes and they are known as tGPI mucins [114]. Structurally, the principal difference between the TcMUC I and II members lies in the central region of the proteins. TcMUC I genes feature a short hypervariable (HV) section and numerous tandem repeats of the Thr_8_-Lys-Pro_2_ sequence, although different variations in this consensus sequence have been observed in the *T. cruzi* genome assemblies with NGS methods. In contrast, TcMUC II genes possess a central region with a longer HV section and fewer tandem repeats, which are still rich in threonine and proline. Some studies have suggested that TcMUC II genes may have evolved from TcMUC I genes or vice versa [139]. Also, TcMUCII members exhibit differential expressions among strains of O-linked α-galactosyl residues, which are crucial for inducing anti–α-Gal antibodies. Variations in their structures likely have significant implications for the diagnosis and immunopathology of Chagas disease, depending on the strain [140].

Besides, TcMUC group III is constituted by a single-copy gene of a mucin-like protein known as TSSA (Trypomastigote Small Surface Antigen), which is only present in the bloodstream of trypomastigote membranes and is not a sialic acid acceptor. TSSA appears to play a role in host cell invasion as an adhesion molecule [141]. Unlike TcMUC I and II genes, it does not display a specific Thr-rich region, and there is a debate about their sequence composition. While a sequence analysis has shown a high content of serine and threonine residues for O-glycosylation [142], another study described TSSA as a rather lowly glycosylated molecule [143].

TcSMUGs (*T. cruzi* small mucin-like genes) consist of two groups of genes, designated as L (large) and S (small), which exhibit over 80% sequence identity, but differ in their genomic structure. TcSMUG S genes form the backbone of GP35/50 mucins, which are expressed during the insect-dwelling stages [144]. In metacyclic trypomastigotes, some researchers have shown that GP35/50 mucins bind to target cells, inducing a bidirectional Ca^2+^ response that facilitates cell invasion [145], while others have suggested that mucin-mediated invasion is accomplished through interaction with host cell annexin A2 and clathrin-dependent endocytosis [146]. Conversely, in epimastigotes, GP35/50 mucins play a protective role against proteases in the insect intestinal tract [147]. Furthermore, TcSMUG S genes, unlike TcSMUG L genes, accept sialic acid residues transferred to the parasite membrane by TSs. Hence, TcSMUG L products are likely involved in attachment to the luminal midgut surface of the insects and are exclusive to the epimastigote stage [148,149].

### 4.4. DGF-1

The disperse gene family 1 (DGF-1) is the fifth largest multi-gene family of *T. cruzi*, and it was placed among the highest expressed proteins in this parasite [150]. Its composition includes the presence of eight to nine transmembrane hydrophobic helices at their C-terminal end, potentially serving as membrane anchors. These genes also exhibit six epidermal growth factor-1-like (EGF-1) domains and one EGF-2-like domain, spaced approximately 400 amino acids apart, along with lectin-binding motifs. Furthermore, integrin-like sequences suggest a cell surface localization, implying roles in cell–cell interactions or signal transduction. Finally, they lack a GPI anchor and around half of the members display a canonical signal peptide [151,152].

Based on a neighbor-joining (NJ) tree and the distribution of functional domains in the CL Brener strain, DFG-1 members were categorized into two principal groups (A and B), which included most of the proteins. However, the rest of the members were clustered in other small groups (C–E) [152]. Therefore, as we explained before for the other multi-gene families, more studies are mandatory in different strains to confirm this classification. Also, the NJ method for phylogeny analysis is less accurate than maximum likelihood or maximum parsimony methods for the calculation of phylogenies [153]. Additionally, phylogenetic analysis has suggested that in the DGF-1 family, the high sequence similarity among members indicates that these events are either recent or subjected to homogenization processes [152]. Interestingly, DGF-1 copies have been identified in the sub-telomeric regions of BAC-telomere recombinants of the CL Brener strain, and these DGF-1 copies are flanked by trans-sialidases and RHS elements [75]. These sub-telomeric localization of DGF-1 has been confirmed across various *T. cruzi* strains, with significant variation in copy numbers among strains [154].

Regarding the possible roles of DGF-1 proteins in the life cycle of *T. cruzi*, different homology, glycoproteomics, and structural analyses suggest that they function as receptors or bidirectional signal transducers, potentially regulated by cyclic AMP (cAMP), nucleoside analogs, and non-cAMP kinases [154,155,156]. These findings, along with in silico predictions, underscore the potential of DGF-1 genes to interact with other proteins, with these interactions being subject to regulatory mechanisms. However, despite the extensive data yielded by new omics technologies and advancements in immunology, the specific functions of this prominent and ubiquitous protein family remain elusive.

## 5. Conclusions

*T. cruzi* exhibits significant genomic complexity and extensive genetic diversity among strains, as demonstrated by the analyses of multiple strains and the differences observed among them at both the sequence and infection levels. Its genome is specifically organized into two distinct compartments, the core and the disruptive, which display different gene composition. Research into this genetic diversity has yielded vital insights into the parasite pathogenicity, and the complex array of genetic variations within *T. cruzi* populations poses key challenges in developing effective vaccines and therapies for Chagas disease. Notably, advances in NGS technologies, with the integration of long- and short-read sequencing methods, have significantly enhanced the quality of the *T. cruzi* genomes, shedding light on this extensive genetic diversity and complexity.

Furthermore, the information described about the different multi-gene families of *T. cruzi* confirms their critical role for the infection processes and parasite survival. These families have undergone significant growth and continuous evolution, enhancing the *T. cruzi* capacity to adapt, survive, and infect both insect and mammalian hosts. Most of the multi-gene families are exclusive to this type of parasites, which makes them conducive to the design of specific vaccines. Functional genomics, omics approaches, new structural and sequence analysis with the most recent assembled genomes and in vivo research may provide comprehensive insights into their biological functions. Altogether, these advances may allow the design of effective new treatments against Chagas disease.

## Figures and Tables

**Figure 1 pathogens-14-00061-f001:**
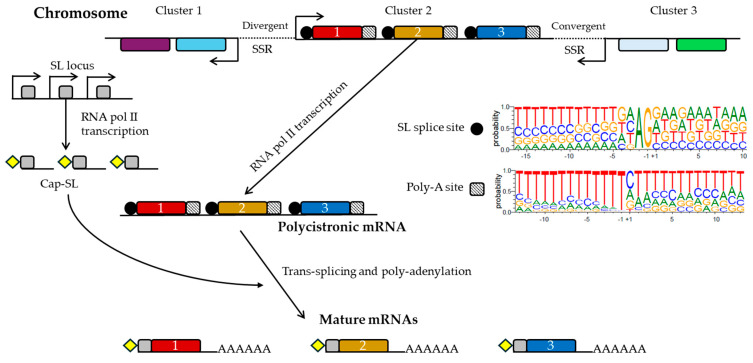
**Polycistronic transcription of *T. cruzi* and processing**. Gene clusters in the genome are transcribed to polycistronic mRNAs by the RNA polymerase II, as displayed by Cluster 2. Divergent and convergent SSRs, as well as the spliced leader (SL) and the poly-A insertion sites with the nucleotide distribution in *T. cruzi* according to Callejas-Hernández et al. [62], are shown. Then, this polycistronic transcript is processed by trans-splicing and polyadenylation to produce the mature and individual mRNAs. For the trans-splicing, a capped SL RNA (grey box) is added to the 5′ end of every mRNA. The SL locus is in a different chromosome and is also transcribed by the RNA polymerase II. Cap modification is shown by a yellow diamond.

**Figure 2 pathogens-14-00061-f002:**
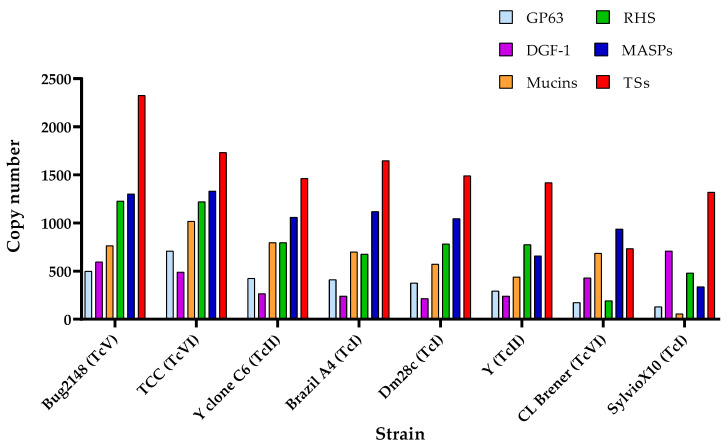
**Copy number of the six principal multi-gene families in different strains of *T. cruzi***. TSs: trans-sialidases. RHS: retrotransposon hot spot proteins. MASPs: mucin-associated surface proteins. DGF-1: disperse gene family 1 members. Based on the data of Berná et al. [55], Callejas-Hernández et al. [47,62], Wang et al. [59], and El-Sayed et al. [42].

**Figure 3 pathogens-14-00061-f003:**
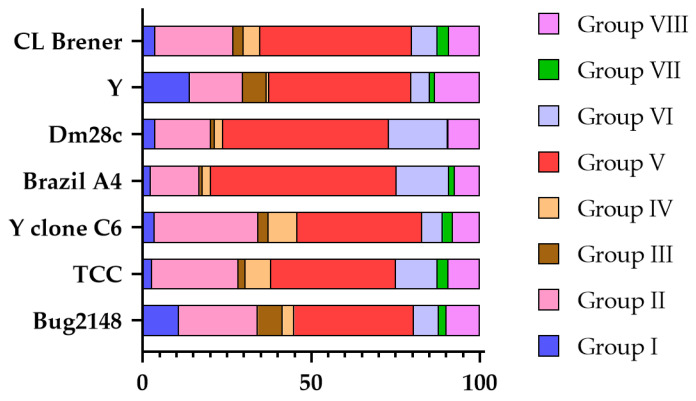
TS group distribution according to the annotated trans-sialidases in each strain of *T. cruzi* based on the data of Berná et al. [55], Callejas-Hernández et al. [47], Freitas et al. [122], and Wang et al. [59]. The percentage of each group is shown with different colors.

**Figure 4 pathogens-14-00061-f004:**
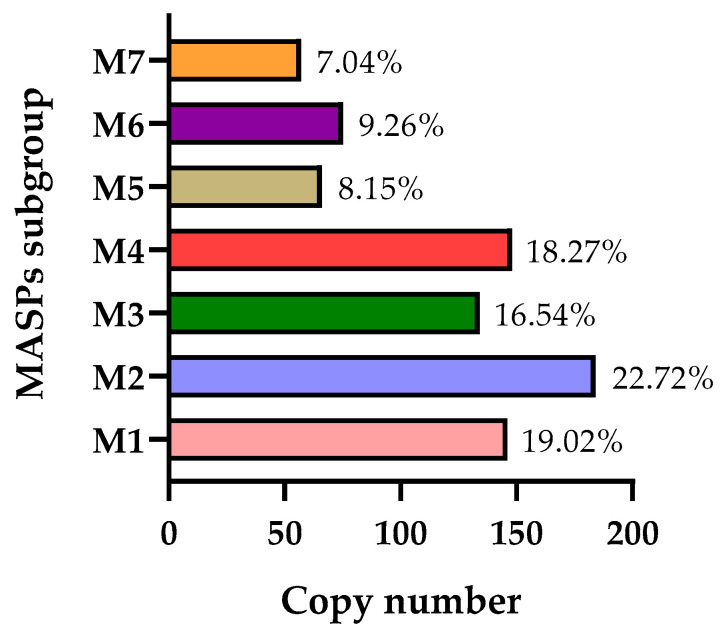
Copy number of each MASP subgroup in the CL Brener strain. The percentage of each subgroup is displayed.

**Figure 5 pathogens-14-00061-f005:**
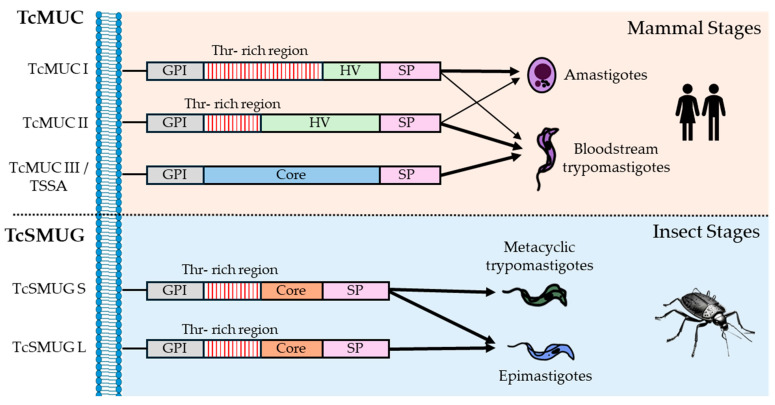
**Mucin classification**. Schematic representation of each type of mucin is displayed, as well as the life cycle stages in which they are expressed. The width of the arrow corresponds to the expression levels in each stage. SP: signal peptide. HV: hypervariable region.

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
