# Peer review of "Trypanosoma cruzi*: Genomic Diversity and Structure"

_pathogens, 2025, doi:10.3390/pathogens14010061_

Round 1

Reviewer 1 Report

Comments and Suggestions for Authors

Manuscript is a relevant review of Trypanosoma cruzi. It will certainly contribute to the field, since it compiles genome structure and function – replication and transcription. Advances in the field as well as limitations that need to be considered in data interpretation is discussed. To my knowledge, references are cited properly. I have only some format questions that I believe could improve the rational of the text:

In two different paragraphs, authors discuss T. cruzi sexual reproduction: lines 48-55 and lines 84-91. All considerations concerning this subject should be put together.

The same for DTUs: it appears in lines 76-83 and 107-120. Text should be restructured in order to discuss all considerations together.

Line 115: text says in “2021 another research… “ but the reference is not there.

Lines: 272-280: Authors are discussing a hypothesis that was suggested in the Araujo et al paper concerning genetic variability as consequence of collisions. Therefore, I suggest that instead of can and leads, authors should use: could and would lead.

I couldn’t find Figure 1 cited in the text.

Line 363: Authors say: Several authors, but there is just one reference (48) cited.

It would be better if authors include the references that allowed the generation of figures 2 and 3.

It is not clear what means the immune system pressures they encounter in mammalian hosts (Line 508). Why different proteins in the cell surface encounter different pressures? I couldn’t find this explanation in the ref 40 cited for this sentence.

Finally, sequencing is not a good option as keyword.

Author Response

Reviewer 1

Manuscript is a relevant review of Trypanosoma cruzi. It will certainly contribute to the field, since it compiles genome structure and function – replication and transcription. Advances in the field as well as limitations that need to be considered in data interpretation is discussed. To my knowledge, references are cited properly. I have only some format questions that I believe could improve the rational of the text:

We thank the reviewer for his/her comments and suggestions that will improve the manuscript.

Comment 1. In two different paragraphs, authors discuss T. cruzi sexual reproduction: lines 48-55 and lines 84-91. All considerations concerning this subject should be put together. The same for DTUs: it appears in lines 76-83 and 107-120. Text should be restructured in order to discuss all considerations together.

We agree with the reviewer. We have put together both sections. Lines 48-62 for the T. cruzi sexual reproduction and lines 104-123 for DTUs section.

Comment 2. Line 115: text says in “2021 another research… “ but the reference is not there.

We have added the reference in that sentence (line 121).

Comment 3. Lines: 272-280: Authors are discussing a hypothesis that was suggested in the Araujo et al paper concerning genetic variability as consequence of collisions. Therefore, I suggest that instead of can and leads, authors should use: could and would lead.

We have changed both words in lines 285 and 287.

Comment 4. I couldn’t find Figure 1 cited in the text.

Figure 1 is cited in line 311.

Comment 5. Line 363: Authors say: Several authors, but there is just one reference (48) cited.

We have indicated the name of the first author of that reference (Berná et al.) in line 379.

Comment 6. It would be better if authors include the references that allowed the generation of figures 2 and 3.

We have included those references in both figures.

Comment 7. It is not clear what means the immune system pressures they encounter in mammalian hosts (Line 508). Why different proteins in the cell surface encounter different pressures? I couldn’t find this explanation in the ref 40 cited for this sentence.

We agree with the reviewer. The correct reference, mentioned in that section in lines 521 and 528, is Buscaglia et al., 2006 (reference 138). For example, in this article the authors hypothesize that the diversity of TcMUC compared to TcSMUG “it is certain to be partly related to their distinct expression profile, which puts them under different selective pressures” among other explanations. We have reordered the ideas in lines 526-528 and correctly attributed each reference to address this issue.

Comment 8. Finally, sequencing is not a good option as keyword.

 We have deleted it from the keywords.

Reviewer 2 Report

Comments and Suggestions for Authors

Overall, the manuscript is a good review. I congratulate the authors in providing the contemporary revision.

I have a few comments and suggestions to improve the manuscript, which are as follows:

·      Given the title of the manuscript, I believe it will greatly improve if actual numbers are given in terms to describe the genetic diversity between strains of T. cruzi. For example, simply stating that it is diverse, and that diversity exists between strains, does not allow comparisons to other parasites. My suggestion would be to try to include a couple of sentences in the manuscript that quantitative compare the diversity between strains (e.g., TcI has been described to be 2% divergent to TcII? Or 5%?, please provide values).

·      Line 9. Change “agent of the Chagas” to “agent of Chagas”

·      Line 21. Change “glycosylated proteins on” to “glycosylated proteins expressed on”

·      Line 32. Change “It is the case” to “In the case of Trypanosoma cruzi, this parasite is responsible”

·      Line 49. It is my belief that those references (7 & 8) were not the first to report the hybridization evidence in T. cruzi. If so, please include original studies. If I am incorrect, please disregard my comment.

·      Line 56. I understand the authors want to make the point that genetic recombination through sex has been recorded in T. cruzi, however, I think it’s important to underscore that the main form of reproduction in T. cruzi is clonal.

·      Line 72. I suggest adding a sentence that briefly mentions the biological role or advantage the post-transcriptional edition has in T. cruzi. Many studies often cite this particularity in T. cruzi, but almost none of the studies mention the reason behind this post-transcriptional edition (e.g., is it advantageous?), thus it would be nice if this review mentions something about this.

·      Line 78. A recent study (“Phylogenetic diversity of two common Trypanosoma cruzi lineages in the Southwestern United States. https://doi.org/10.1016/j.meegid.2022.105251”) also described what appears to be a clade that is equally divergent as if it were another DTU. I recommend the authors include this study, and maybe clarify that as additional vectors, hosts and geographical areas are explored, additional lineages (i.e., DTUs) could be found and described.

·      Line 81. How different are those six thousand strains though? Did they all cluster within the current DTUs? Maybe just clarify this, to avoid confusion with the audience that is not familiar with T. cruzi genetics and might incorrectly infer that there are potentially thousands of new DTUs.

·      Line 81. “The wide variety”. This is a good place were a quantitative value of the differences between strains or DTUs could be mentioned, that would allow comparison with other pathogens. See my first comment.

·      Line 88. “excess of heterozygosity”. Rather than this being evidence of a sexual cycle, the excess heterozygosity is what is expected to occur when there is an absence of recombination in a clonal diploid genome, since each homologous chromosome will start to accumulate its own particular mutations. This phenomenon is explained in the introduction of the Machado and Ayala paper of 2001 (www.pnas.orgycgiydoiy10.1073ypnas.121187198 ). These authors in turn cite: 1) White, M. J. D. (1945) Animal Cytology and Evolution (Cambridge Univ. Press, Cambridge, U.K.).;  2) Suomalainen, E. (1950) Adv. Genet. 3, 193–252. and; 3)Lokki, J. (1976) Hereditas 83, 57– 64.

·      Line 91. The citation 25 was not the first to cite the existence of a sexual cycle in T. cruzi. Authors should cite Machado and Ayala 2001 and Gaunt M.W. et al. 2003.

·      Lines 115-120 need a citation.

·      Line 129. Try to avoid using the word “first” two times in the same sentence.

·      Line 140. I would suggest adding in parenthesis the DTU of each strain when it is mentioned in the manuscript (not only in these lines, but throughout the manuscript). This will be very helpful for anyone reading the article.

·      Line 164-165. Include a citation for the synteny reference.

·      Line 169. Include references that confirm that TcI and TcII are the most frequent strains found.

·      Line 182. Change “its” to “is”

·      If possible, keep table 1 in the same page, to facilitate the interpretation of the table, since column titles cannot be read in the next page.

·      Line 225. It is not clear to me if the “gene expansion” is facilitated through errors during homologous recombination events, which can lead to gene duplications in one strand and gene deletions in the other strand. Since in my opinion this is a very interesting possibility, it would be great if the authors can clarify how (i.e., what mechanism) the “gene expansion” would be facilitated.

·      Line 236. “contains highly conserved genes”. Please include citation.

·      Line 286. When authors talk about the absence of a nucleolus and abundant heterochromatin in trypomastigotes is evidence of “transcriptional inactivity”. If so, please clarify.

·      Line 365. Is it known why RHSm GP63 and DGF-1 are found in both disruptive and core compartments? This is simply my curiosity. If it is known or not known, I think it should be mentioned.

·      Figure 2. Mention the DTU of each strain in figure.

·      Lines 409-411. I assume the TS are not only targeted by human anti-alpha-galactosyl antibodies, but potentially by all other hosts T. cruzi infects as well. If so, it might be worth clarifying this in that sentence.

·      Line 4141. It is not clear from the sentence that mentions 5%, that the idea is that of all TS that have been described, only 5% of those gene-copies are expected to have that function. Or if they mean that only 5% of the proteins have that active site. I assume they mean the first, however I recommend clarifying this.

·      Line 470. Clarify why “unspecific”. I assume they mean it’s because T. cruzi is expressing many copies at the same time, but the sentence will benefit if this is expressed more clearly.

·      Line 506. Maybe change the order in which the ideas are expressed, since I would think that its due to the fact that the proteins are under constant pressure (i.e., natural selection) and have therefore been located to their current chromosomal location.

·      Paragraph of lines 527-534 would benefit if an average is provided in terms of the percentage each TcMUC subfamily is composed of in terms of all TcMUCs.

·      Line 569. I would believe that not only DGF-1 gene copies were created by gene duplications, but that all protein families are usually originated by gene duplications.

Author Response

Reviewer 2:

Overall, the manuscript is a good review. I congratulate the authors in providing the contemporary revision. 

I have a few comments and suggestions to improve the manuscript, which are as follows:

 We thank the reviewer for his/he rcomments , corrections and suggestions that will improve the manuscript.

  • Given the title of the manuscript, I believe it will greatly improve if actual numbers are given in terms to describe the genetic diversity between strains of T. cruzi. For example, simply stating that it is diverse, and that diversity exists between strains, does not allow comparisons to other parasites. My suggestion would be to try to include a couple of sentences in the manuscript that quantitative compare the diversity between strains (e.g., TcI has been described to be 2% divergent to TcII? Or 5%?, please provide values).

Thank you for your suggestion. We have added two quantitative comparisons about the DNA content among strains in lines 84-86 together with their references (26 and 27).

Also, we believe that the multi-gene families data presented in the manuscript (Figures 2 and 3), together Tables 1 and 2 can serve for this purpose to quantitatively compare the diversity among strains. 

  • Line 9. Change “agent of the Chagas” to “agent of Chagas”

We have changed it. Line 9. 

  • Line 21. Change “glycosylated proteins on” to “glycosylated proteins expressed on”

We have changed it. Line 21.

  • Line 32. Change “It is the case” to “In the case of Trypanosoma cruzi, this parasite is responsible”

We have changed it. Line 32.

  • Line 49. It is my belief that those references (7 & 8) were not the first to report the hybridization evidence in T. cruzi. If so, please include original studies. If I am incorrect, please disregard my comment.

Thank you for your suggestion. Indeed, they are not the first, although they are some of the more relevant. We have included 4 older studies that reported it. References 9-12 (line 50).

  • Line 56. I understand the authors want to make the point that genetic recombination through sex has been recorded in T. cruzi, however, I think it’s important to underscore that the main form of reproduction in T. cruzi is clonal.

We agree with the reviewer, and we have highlighted this fact in lines 56-57.

  • Line 72. I suggest adding a sentence that briefly mentions the biological role or advantage the post-transcriptional edition has in T. cruzi. Many studies often cite this particularity in T. cruzi, but almost none of the studies mention the reason behind this post-transcriptional edition (e.g., is it advantageous?), thus it would be nice if this review mentions something about this.

We have added a couple of sentences in lines 305-309 and a new reference (95). We think that it is more appropriate to include this section in this part of the review, when we describe the transcription process in the parasite.

  • Line 78. A recent study (“Phylogenetic diversity of two common Trypanosoma cruzi lineages in the Southwestern United States. https://doi.org/10.1016/j.meegid.2022.105251”) also described what appears to be a clade that is equally divergent as if it were another DTU. I recommend the authors include this study, and maybe clarify that as additional vectors, hosts and geographical areas are explored, additional lineages (i.e., DTUs) could be found and described.

We have added this reference and an explanation following your recommendation in lines 110-113.

  • Line 81. How different are those six thousand strains though? Did they all cluster within the current DTUs? Maybe just clarify this, to avoid confusion with the audience that is not familiar with T. cruzi genetics and might incorrectly infer that there are potentially thousands of new DTUs.

To avoid this misunderstanding, we have deleted the previous sentence about the DTUs (line 83), as we described the DTUs for the first time and more extensively in the following section of the review. We think that now the sentence is correct. Just in case and answering your question, all those six thousand strains belong to the distinct DTUs.

  • Line 81. “The wide variety”. This is a good place where a quantitative value of the differences between strains or DTUs could be mentioned, that would allow comparison with other pathogens. See my first comment.

We included the information of your first comment at this point.

  • Line 88. “excess of heterozygosity”. Rather than this being evidence of a sexual cycle, the excess heterozygosity is what is expected to occur when there is an absence of recombination in a clonal diploid genome, since each homologous chromosome will start to accumulate its own particular mutations. This phenomenon is explained in the introduction of the Machado and Ayala paper of 2001 (www.pnas.orgycgiydoiy10.1073ypnas.121187198 ). These authors in turn cite: 1) White, M. J. D. (1945) Animal Cytology and Evolution (Cambridge Univ. Press, Cambridge, U.K.).;  2) Suomalainen, E. (1950) Adv. Genet. 3, 193–252. and; 3)Lokki, J. (1976) Hereditas 83, 57– 64.

We agree with the reviewer, and we have deleted this part.

  • Line 91. The citation 25 was not the first to cite the existence of a sexual cycle in T. cruzi. Authors should cite Machado and Ayala 2001 and Gaunt M.W. et al. 2003.

We have added them to this point (references 9 and 17, line 62).

  • Lines 115-120 need a citation.

We have added that reference (41) to line 121.

  • Line 129. Try to avoid using the word “first” two times in the same sentence.

We have deleted one of the “first”. Line 132.

  • Line 140. I would suggest adding in parenthesis the DTU of each strain when it is mentioned in the manuscript (not only in these lines, but throughout the manuscript). This will be very helpful for anyone reading the article.

The DTU of each strain is already indicated in a specific column in Tables 1 and 2. We believe that repeatedly clarifying the DTU throughout the text may disrupt the reading flow of this review.

  • Line 164-165. Include a citation for the synteny reference.

We agree with the reviewer, and we have included a reference (56) for this sentence (line 168).

  • Line 169. Include references that confirm that TcI and TcII are the most frequent strains found.

We realized that this sentence was not accurate. For example, depending on the geographical area, each DTU is more frequent than the others (DOI: https://doi.org/10.3390/life13122339). Therefore, we have deleted that sentence.

  • Line 182. Change “its” to “is”

We have changed it (line 185).

  • If possible, keep table 1 in the same page, to facilitate the interpretation of the table, since column titles cannot be read in the next page.

We agree with the reviewer. Table 1 was originally on the same page, but due to the manuscript template and format of the journal it was changed. We will talk to the editor to manage this option.

  • Line 225. It is not clear to me if the “gene expansion” is facilitated through errors during homologous recombination events, which can lead to gene duplications in one strand and gene deletions in the other strand. Since in my opinion this is a very interesting possibility, it would be great if the authors can clarify how (i.e., what mechanism) the “gene expansion” would be facilitated.

To the best of our knowledge, just a few mechanisms have been proposed to describe the 'gene expansion' in T. cruzi. The main mechanism consists of homologous recombination at the disruptive compartment of the genome, mainly at the telomeric and subtelomeric regions. This mechanism includes the participation of transposable elements' nucleases, which introduce chromosome breakages, and the subsequent repair (by homologous recombination) introduces new gene copies, variants, and pseudogenes. This mechanism also considers (in a lesser proportion) the possibility of recombination between non-homologous chromosomes (ectopic recombination).

We have added this information and the associated references in lines 234-241.

  • Line 236. “contains highly conserved genes”. Please include citation.

The citation is at the end of the paragraph, as all the explanation is based on the same references (78 and 79, line 251).

  • Line 286. When authors talk about the absence of a nucleolus and abundant heterochromatin in trypomastigotes is evidence of “transcriptional inactivity”. If so, please clarify.

We agree with the reviewer, and we have added that in line 296.

  • Line 365. Is it known why RHS, GP63 and DGF-1 are found in both disruptive and core compartments? This is simply my curiosity. If it is known or not known, I think it should be mentioned.

As far as we know, no study has clarified the question you raise. Following your suggestion, we have included a sentence indicating this (line 382).  

  • Figure 2. Mention the DTU of each strain in figure.

We have mentioned DTUs in the figure (line 405).

  • Lines 409-411. I assume the TS are not only targeted by human anti-alpha-galactosyl antibodies, but potentially by all other hosts T. cruzi infects as well. If so, it might be worth clarifying this in that sentence.

We respectfully think that the reviewer has misunderstood this part. TSs are not targeted by these antibodies; the targets are the glycoproteins (i.e. alpha-galactosylmucins) of the parasite membrane. When these glycoproteins accept the sialic acids transferred by the TSs, they are protected against the action of the anti-alpha-galactosyl antibodies. This is explained in lines 424-429.

  • Line 414. It is not clear from the sentence that mentions 5%, that the idea is that of all TS that have been described, only 5% of those gene-copies are expected to have that function. Or if they mean that only 5% of the proteins have that active site. I assume they mean the first, however I recommend clarifying this.

Thank you for asking this question. After reviewing various references in this field, we realized that none of them answer your doubt, and the original publication is not publicly available. Therefore, we have decided to delete that sentence as we cannot provide an accurate answer.

  • Line 470. Clarify why “unspecific”. I assume they mean it’s because T. cruzi is expressing many copies at the same time, but the sentence will benefit if this is expressed more clearly.

We have rewritten this section to express it more accurately (lines 488-491).

  • Line 506. Maybe change the order in which the ideas are expressed, since I would think that its due to the fact that the proteins are under constant pressure (i.e., natural selection) and have therefore been located to their current chromosomal location.

We have changed the order of the expressed ideas (lines 526-528).

  • Paragraph of lines 527-534 would benefit if an average is provided in terms of the percentage each TcMUC subfamily is composed of in terms of all TcMUCs.

We agree with the reviewer, but currently this point cannot be assessed. Unfortunately, T. cruzi assemblies label several mucin genes merely as putative TcMUC mucins or just as putative mucin-like proteins, which prevents us from determining the proportion of each TcMUC subfamily. Therefore, we could not accurately comment on this

  • Line 569. I would believe that not only DGF-1 gene copies were created by gene duplications, but that all protein families are usually originated by gene duplications.

We agree with the reviewer, and we have eliminated that sentence.